# Use of Letermovir for CMV Prophylaxis after Allogeneic Hematopoietic Stem Cell Transplantation: Review of the Literature and Single-Center Real-Life Experience

Jessica Gill [1,2], Davide Stella [1,2], Irene Dogliotti [1], Chiara Dellacasa [1], Luisa Giaccone [1,2,†] and Alessandro Busca [1,*,†]

1   Department of Oncology and Hematology, SSD Stem Cell Transplant Center, A.O.U. Città della Salute e della Scienza di Torino, 10126 Turin, Italy; jgill@cittadellasalute.to.it (J.G.)
2   Division of Hematology, Department of Molecular Biotechnology and Health Sciences, University of Torino, 10126 Turin, Italy
*   Correspondence: abusca@cittadellasalute.to.it
†   These authors contributed equally to this work.

**Abstract:** Cytomegalovirus (CMV) reactivation after allogeneic hematopoietic stem cell transplant (allo-HSCT) is mainly due to an increase of latent viremia in previously exposed patients. Furthermore, CMV reactivation in this setting has a significant impact on patient survival. Traditional approach to CMV reactivation post allo-HSCT was a pre-emptive treatment with antivirals in the case of increased viremia. However, since 2017, a new antiviral compound, letermovir, has been introduced in clinical practice and is deeply changing the common CMV approach. The toxicity profile of letermovir allowed its use in prophylaxes in patients at high risk of CMV reactivation. This review will focus on the present role of letermovir post allo-HSCT and discuss some possible future applications of the drug. Finally, our single center CMV management in view of the recent introduction of letermovir will be discussed.

**Keywords:** allogeneic hematopoietic stem cell transplant; CMV; pre-emptive therapy; letermovir

## 1. Introduction

Cytomegalovirus (CMV) infection remains a significant cause of morbidity and mortality after allogeneic hematopoietic stem cell transplantation (allo-HSCT) [1,2]. CMV replication occurs in 50–60% of these patients either as a reactivation of a latent viremia or as a new infection. Without any prophylaxis or pre-emptive therapy (PET), 20% to 30% of them may develop end-organ disease [3]. Furthermore, CMV infection predisposes patients to several indirect effects, such as Graft-Versus-Host-Disease (GVHD) [4], myelosuppression and invasive bacterial and fungal infection [5,6], leading to increased non-relapse mortality (NRM) [7–11]. However, several studies have suggested that CMV replication could enhance leukemia-specific T cells, eventually reducing the risk of relapse [12].

Several risk factors for CMV infection (CMV-i) have been recognized, mainly regarding the host serological status and transplant characteristics [13]. CMV seropositive recipients are at higher risk for CMV reactivation, whereas the role of donor serum status is more controversial [14–16]. Moreover, the risk of CMV reactivation is enhanced with: (1) myeloablative conditioning, which ablates any pre-existing CMV-specific T-cell immunity [17–19]; (2) the use of ex vivo T-cell-depleted grafts or umbilical cord blood, in which pathogen-specific immunity is not transferred [16,20]; (3) GVHD prophylaxis with ATG or post-transplantation cyclophosphamide, as this causes absolute lymphopenia after transplant, with a low CD4+ T cell count and undetectable CMV-specific T-cell immunity [21]; and (4) the presence of GVHD, requiring treatment with prolonged calcineurin inhibitors (CNI) or high-dose steroids [1].

Current standard-of-care therapies for CMV-i are typically intravenous ganciclovir or its oral prodrug, valganciclovir, whereas intravenous foscarnet and cidofovir are considered as second line agents for the treatment of resistant and refractory CMV. Novel immunotherapeutic approaches with the infusion of vaccines or adoptive CMV-specific lymphocytes have been tested to treat CMV infections that are unresponsive to antiviral therapy with promising results [22]. The use of CMV-specific immunoglobulins is still controversial [23].

Many studies have demonstrated that a measurable DNAemia is an important risk factor for the development of CMV disease and is a relevant prognostic indicator [24]. A pivotal role of PET is, indeed, crucial, based on strict surveillance of the CMV viremia and prompt start of antiviral therapy when a target threshold of the viral load is reached. The final purpose is to prevent clinical CMV disease and minimize the toxic effects of antiviral agents [25].

However, this strategy is poorly standardized: infection can be detected in the blood or plasma using different methods either with CMV antigenemia or more frequently with viral DNAemia detected by PCR. The frequency of monitoring is typically once or twice weekly. The optimal CMV level threshold to start therapy remains to be defined, depending on host's risk for CMV disease, current immunosuppression and treatment center practice, as suggested by ECIL-7 guidelines [23]. Several randomized studies have shown that this strategy efficiently decreased the incidence of early CMV disease after allo-HSCT from 25–30% to <5% [26–29]. Nevertheless, CMV reactivation remains associated with a poor clinical outcome [12].

A different strategy against CMV is based on the prophylactic use of antiviral drugs post-HSCT in the absence of detectable CMV DNAemia [30–34]. The main advantage of prophylaxis is the prevention of viral replication in the early post-transplant period, reducing potential CMV-related effects that could increase NRM. Despite their efficacy, the prophylactic use of foscarnet, ganciclovir and cidofovir in prophylaxis is restricted by clinically significant toxic effects (especially on bone marrow and renal function) and the potential emergence of drug resistance [35,36].

Some drugs have been studied for CMV prophylaxis in phase II and III clinical trials (maribavir and brincidofovir) [37–39] with disappointing results. Since 2017, letermovir (LTV) is the only approved drug for CMV prophylaxis HSCT recipients at high risk for CMV infection [28].

## 2. Letermovir

LTV is a new anti-CMV agent targeting the viral terminase subunits pUL56 and pUL5 1, which are components of the terminase complex involved in DNA cleavage and packaging of the viral progeny [40,41]. LTV is available as oral tablets and as an intravenous solution. The oral bioavailability is 94%; however, in HSCT, recipients can drop down to 35%. LTV elimination is mainly biliary, involving the hepatic active transport transporters OATP1B1/3 and the UGT1A1/3 for glucuronidation.

LTV is a weak-to-moderate inhibitor of CYP3A4, resulting in increased levels of sirolimus and tacrolimus; a weak-to-moderate inducer of CYP2C9/19 potentially resulting in increased levels of voriconazole; and an inhibitor of OATP1B1/3. The LTV recommended dose is 480 mg once a day. Co-administration with OATP1B1/3 inhibitors may result in increased plasma concentrations of LTV; in this respect, when LTV is co-administered with cyclosporine, a potent OATP1B1/3 inhibitor, the LTV dose should be halved (240 mg once daily) [28,42]. Another frequently used drug that inhibits OATP1B is rifampin.

Overall, LTV is generally well tolerated, with gastrointestinal symptoms, including nausea and diarrhea, as the most common adverse events. Other main side effects reported are cough, headache and fatigue. Nephrotoxicity and myelotoxicity, the most common adverse effects of other anti-CMV drugs, were not observed in patients receiving LTV and included in the phase III clinical trial [28]. The excellent safety profile of LTV is probably due to the lack of a mammalian counterpart to the viral terminase complex.

No dose adjustment of LTV is necessary in the presence of mild (Child–Pugh class A) or moderate (Child–Pugh class B) hepatic impairment, while LTV is not recommended in patients with severe hepatic impairment. Similarly, dose adjustment of LTV is not required in patients with mild, moderate or severe renal impairment [43].

CMV genotyping in patients with clinically significant CMV infection (CS-CMV-i) during LTV therapy, revealed mainly UL56 gene mutations associated with reduced susceptibility to LTV, in particular V236M and C325W mutant during LTV prophylaxis and E237G and R369T mutant after its discontinuation; in these cases, patients were rescued by ganciclovir [44].

## 3. Pre-Emptive Therapy Versus Letermovir Prophylaxis

Many studies have compared the two CMV approaches: PET versus LTV prophylaxis administered up to day 100 post HSCT (Table 1).

**Table 1.** Main studies comparing placebo and/or PET versus LTV prophylaxis since 2014.

| STUDY | n | CS-CMV-i at 3 Mo | | | All-Cause M at 6 Mo | | |
|---|---|---|---|---|---|---|---|
| | (Control */LTV) | Control (%) | LTV (%) | *p*-Value | Control (%) | LTV (%) | *p*-Value |
| PROSPECTIVE: vs. Placebo * | | | | | | | |
| Chemaly et al. [45], 2014 | 33/34 | 36 | 6 | 0.007 | - | - | - |
| Marty et al. [28], 2017 | 170/325 | 50 | 19.1 | <0.001 | 15.9 | 10.2 | 0.03 |
| Ljungman et al. [46], 2020 | 170/325 | - | - | - | 17.2 | 12.1 | 0.04 |
| Marty et al. [47], 2019 | 22/48 | 90.9 | 45.8 | <0.001 | 18.2 | 15 | 0.268 |
| RETROSPECTIVE: vs. PET * | | | | | | | |
| Malagola et al. [48], 2020 | 41/45 | 44 | 8 | 0.0006 | - | - | - |
| Anderson et al. [49], 2020 | 106/25 | 59 | 4 | <0.001 | 48 | 44 | 0.27 |
| Studer et al. [50], 2020 [†] | 353/28 | 24.6 | 7.1 | <0.003 | - | - | - |
| Mori et al. [51], 2020 [†] | 571/114 | 71.3 | 20.6 | <0.001 | 14.9 | 8.9 | 0.052 [#] |
| Johnsrud et al. [52], 2020 | 637/108 | 48.8 | 12 | <0.001 | - | - | - |
| Lin et al. [53], 2021 [†] | 32/32 | 68.8 | 21.9 | 0.001 | - | - | - |
| Royston et al. [54], 2021 [†] | 52/26 | 82.7 | 34.6 | <0.0001 | 25 | 15.4 | 0.40 |
| Serio et al. [55], 2021 | 22/13 | 68 | 7.7 | <0.0001 | - | - | - |
| Sperotto et al. [56], 2021 | 55/55 | 60 | 4 | 0.03 | 42 | 25.5 | 0.03 |
| Derigs et al. [57], 2021 | 80/80 | 41 | 14 | <0.001 | - | - | - |
| Wolfe et al. [58], 2021 [†] | 143/119 | 56.6 | 24.4 | <0.001 | 43.4 | 31.9 | 0.06 |
| Yoshimura et al. [59], 2022 [†] | 73/38 | 56.2 | 29.7 | <0.001 | 22.1 | 11 | 0.148 [#] |

* Control is defined by placebo in prospective studies and pre-emptive therapy in retrospective ones. CS-CMV-i is considered as significant CMV reactivation requiring any therapy within 3 months or within 6 months in studies marked by †. Mortality comprises death due to any cause, except in two studies (#). Abbreviations: n—number, CS-CMV-i—clinically significant CMV infection, M—mortality, mo—months, LTV—letermovir, and PET—pre-emptive therapy.

Chemaly et al. [45] first demonstrated that the reduction of virologic failure with LTV is dose-dependent, with the highest anti-CMV activity with the dose of 240 mg per day and an acceptable safety profile. These patients had few CMV reactivations; however, a significantly reduced time to the onset of prophylaxis failure has been reported. A post hoc analysis revealed that these patients had active CMV replication before they received the study drug; excluding them, there were no cases of virologic failure in the group of patients who received 240 mg per day of LTV.

In a prospective randomized study, Marty et al. [28] showed that LTV was able to prevent CS-CMV-i in patients at high and low risk for CMV disease and reduced mortality particularly in high-risk patients at 6 months from allo-HSCT. Later, they investigated outcomes in patients with detectable CMV-DNA at randomization who were excluded from the primary efficacy analysis [47]. LTV resulted as being more effective than placebo in preventing reactivation without any significant effect on mortality. Overall, the clinical

outcomes were similar to those reported in patients with undetectable CMV-DNA at randomization.

A post hoc analysis of a phase III trial [46] evaluated the effect of LTV on all-cause mortality. The Kaplan–Meier rate for all-cause mortality at week 24 post-HCT was lower in the LTV group (12.1%) compared with the placebo group (17.2%; $p = 0.04$); interestingly, the incidence of all-cause mortality in the LTV group was similar in patients with or without CS-CMV-i (15.8 vs. 19.4%; $p = 0.71$).

LTV also significantly improved CMV-related events, such as bacterial and fungal infections, CMV disease and hospital readmission, as demonstrated by Malagola et al. [48]; this result was even more evident at six months after allo-HSCT. Similarly, two studies (Anderson et al. [49] and Studer et al. [50]) suggested a sustained efficacy of LTV on CMV reactivation after discontinuation of prophylaxis 200 days post-HSCT.

Two recent retrospective analysis on large cohorts of patients (Mori et al. [51]; Johnsrud et al. [52]) demonstrated that prophylactic LTV is highly effective in preventing development of CS-CMV-i and ultimately reduces transplant-related mortality (TRM), even in patients with multiple risk factors for CMV reactivation. Moreover, a multivariate analysis demonstrated that LTV prophylaxis is an independent risk factor for a better overall survival (OS) after allo-HSCT.

A handful of studies confirmed the efficacy of primary LTV prophylaxis in preventing CS-CMV-i in very high-risk population, such as haploidentical transplant recipients (Lin et al. [53]) and patients with acute GVHD (Wolfe et al. [58]).

Royston et al. [54] reported a better hematological reconstitution and renal function in patients treated with LTV with more robust platelet recovery. Both these advantages can be attributed to less CS-CMV-i recurrences and overall exposure to valganciclovir treatment, two common causes of renal impairment and post-HSCT cytopenia leading to poor graft function. The improvement on immune reconstitution may be balanced by a potential delay in CMV-specific T-cell reconstitution due to a reduced antigen exposure with antiviral prophylaxis. A recent real-word experience confirmed the efficacy and safety of LTV, with a CS-CMV-i CI at day +100 and +168 of 5.4% and 18.1% and an overall mortality of 6.4% and 7.3%, respectively [60].

Interestingly, Serio et al. [55] demonstrated that the rates of CMV reactivation in patients receiving LTV prophylaxis were similar in autologous and allogeneic transplant recipients (7.7% in allogeneic transplants receiving LTV vs. 68% in allogeneic transplants not receiving LTV vs. 15% in autologous transplants; $p < 0.0001$), without a significant impact on OS.

## 4. CMV Reactivation after Letermovir Prophylaxis

Although many studies demonstrated that LTV is highly effective for the prevention of CS-CMV-i after allo-HSCT, delayed-onset infections after LTV discontinuation are frequently reported, suggesting a potential role for the extension of LTV prophylaxis beyond day 100 [60–64].

In a retrospective analysis, Sperotto et al. [56] showed that CMV reactivation occurred more frequently within 100 days after HSCT in the PET group (87% vs. 9%, $p < 0.01$), while, in the LTV prophylactic group, reactivations were observed mainly after drug discontinuation (13% vs. 91%, $p < 0.01$). However, there was a statically significant difference in the peak of DNAemia with a lower copy number in the LTV group versus PET group.

Liu et al. [65] compared the LTV and PET strategy in a group of 333 patients, showing that the initial improvement observed in the first 3 months in the LTV group disappears afterwards. The multivariate analyses demonstrated that the use of LTV was associated with improved OS and reduced NRM and CMV-related mortality from day 0 to day 99 (HR 0.43, $p$ 0.004; HR 0.50, $p$ 0.03; HR 0.40, $p$ 0.04, respectively) but worse CMV-related mortality afterwards (at day 364, HR 3.19, $p$ 0.01) with a trend toward worse OS and NRM (at day 364, HR 1.28 and 1.68, respectively, $p$ ns). An increased risk of CMV reactivation was associated with serum IgG levels <400 mg/dL at day 100, high-risk HSCT ($p$ 0.004), the

use of post-transplantation cyclophosphamide (PTCy; *p* 0.001) and mismatched-unrelated donors (MMUD; *p* = 0.02).

In a similar study, the prophylactic approach reduced the 180-day cumulative incidence of CS-CMV-i (44.7 vs. 72.4%, *p* < 0.001) and improved the OS rate at 180 days after transplant (80.4 vs. 73.0%, *p* = 0.033) with a trend of lower NRM (8.9 vs. 14.9%, *p* 0.052) [51]. In another retrospective study, a lower incidence of CS-CMV-i was also observed in the LTV group in time-to-event analyses censored at day 200 (20% vs. 59%, *p* 0.0003), although CS-CMV-i doubled in the LTV group between day +100 and +200 post-HSCT, after discontinuation of LTV, hence, raising the importance for continued CMV monitoring until day +200 post-HCT [49].

The optimal duration of LTV prophylaxis remains an unanswered question, especially in high-risk allo-HSCT recipients. Scattered studies reported that late CS-CMV-i was rare if LTV prophylaxis was continued beyond 14 weeks. Based on this rationale, an ongoing phase III randomized double-blind placebo-controlled clinical trial is exploring the efficacy and safety of LTV administrated for 200 days post-transplant in CMV-seropositive allotransplant recipients (Clinicaltrials.gov: NCT03930615). Preliminary results showed a strongly lower CS-CMV-i in these patients (2.8% in LTV group vs. 18.9% in placebo group), confirmed at a longer follow-up until 48 weeks (14.6% in LTV group vs. 20.3% in placebo group), recommending LTV continuation until day 200.

## 5. Letermovir as Secondary Prophylaxis

Given the efficacy of LTV as primary prophylaxis, the drug has been studied as secondary prophylaxis (SP) in patients who experienced at least one CMV episode after allo-HSCT [66]. Regrettably, to date, there are only few data published so far, with scattered case reports [67–71].

In a retrospective study conducted by EBMT [71], LTV as SP was used in 40 patients: 10.1% breakthrough CS-CMV-i were reported; however, all of them were successfully rescued with other antivirals and OS at 120 days was 81.9% (95% CI = 65.7–90.9; 7/40 events). A French compassionate program reported the results of 80 CMV-seropositive patients who received LTV as SP [70]. Four patients developed CMV breakthrough infections (n = 1) or disease (n = 3) after the initiation of LTV, and three of these patients carried CMV UL56 mutation C325Y or W, conferring the high-level LTV resistance. Six deaths were reported, but only two were related to CMV infection among other causes.

Based on these data, LTV warrants further investigations as a potential tool for SP.

## 6. Letermovir as Therapy

LTV has also been studied as a treatment in non-responders to standard anti-CMV treatment, and some case reports support the potential efficacy in clearing CMV-DNA and treating CMV disease [71,72]. Interestingly, LTV showed efficacy in UL54 mutant CMV, which is a recurrent mutation found in patients resistant to foscarnet, ganciclovir and cidofovir [73,74].

Schubert et al. [75] studied the effect of LTV in patients refractory to ganciclovir or foscarnet. Nine patients received LTV for CMV-i after allo-HSCT, and all but two responded. The median duration of LTV treatment was 31 (8–127) days. The median treatment duration to achieve a decrease of viral load <200 IU/mL was 23 (8–83) days. All patients received LTV as a monotherapy. The treatment was generally well tolerated, and no adverse events were reported.

In the EBMT study [71], only two patients received LTV to treat CMV pneumonia and CMV colitis, both healed the disease after 35 and 91 days, respectively. However, no antiviral effect was documented in patients treated with LTV as PET (n = 5).

Therefore, LTV may be considered in patients with multidrug-resistant CMV disease or in those intolerant or resistant to current therapies, owing to its favorable side-effect profile with less bone marrow toxicity compared with valganciclovir and ganciclovir and less nephrotoxicity compared with cidofovir and foscarnet.

## 7. Letermovir and Immune Reconstitution

Immune reconstitution after allo-HSCT depends on the use of high-dose steroids, lymphopenia, chronic GVHD and the stem cell source, but it is also deeply influenced by CMV serostatus and CMV replication [76–78]. Decreased CMV-specific cellular immunity at 3 months after allo-HSCT is associated with increased late CMV reactivation and mortality [79]. Ganciclovir prophylaxis, leading to decreased CMV reactivation and reduced antigen exposure, has been shown to play a key role in delayed immune reconstitution [80]. Similarly, decreased CMV reactivation on LTV prophylaxis provides a potential mechanism for the observed increase of CS-CMV-i after LTV discontinuation.

Zamora et al. studied CMV-specific T cell responses at 3 months after allo-HSCT following stimulation with CMV immediate early-1 (IE-1) antigen and phosphoprotein 65 (pp65) antigens with thirteen-color flow cytometry. The study showed that LTV prophylaxis appeared to be associated with delayed polyfunctional CMV-specific cellular immune reconstitution at 3 months compared with PET, and polyfunctional T cell response remained diminished also after adjustment for donor CMV serostatus, absolute lymphocyte count and steroid use.

Among LTV recipients, greater peaks of CMV-DNA and increased viral shedding were associated with stronger CD8+ responses to pp65, whereas the CMV shedding rate was associated with greater CD4+ responses to IE-1 [64]. A different study evaluated the number of absolute numbers of CD4+ and CD8+ T cells in two groups of patients receiving PET or prophylactic LTV. No difference was observed between PET versus LTV in CD4+ and CD8+ T cell recovery at early timepoints. However, a statistically significant difference was detected at day +60 (CD4+: median 270/mL vs. 130/mL, $p = 0.04$; CD8+: median 260/mL vs. 100/mL, $p$ 0.03) and at day +90 (CD4+: median 430/mL vs. 190/mL, $p$ 0.03; CD8+: median 410/mL vs. 270/mL, $p$ 0.04).

After day +180, a progressive increase of immune recovery was observed in the LTV group but the difference between the two groups disappeared at 1 year from transplant only (PET vs. LTV group: CD4: median 650/mL vs. 510/mL, $p$ 0.5; CD8: median 310/mL vs. 290/mL, $p$ 0.4) [56]. The analysis of CMV-specific immunity by intracellular cytokine staining (ICS) and ELISPOT assays in PET and LTV groups, showed a delayed recovery in the latter group (ICS assay: median, 177 days for PET vs. 275 days for LTV; $p$ 0.001; ELISPOT assay: 183 days vs. 307 days; $p$ 0.002) [62].

The same study reported other two interesting observations: (1) time to recovery of CMV-specific immunity was not influenced by the occurrence of abortive infection during LTV prophylaxis; and (2) in both groups, no CS-CMV-i was observed, once protective immunity was achieved, especially if detected by ELISPOT assay [62].

A standardized method to measure post-allotransplant CMV-specific immune recovery might be of great benefit to help a patient tailored management of LTV prophylaxis. In the kidney transplant setting quantiferon CMV positive at day +100 is strongly associated with protection from CS-CMV-i after this time. In allo-HSCT setting, a landmark analysis demonstrated that IgG level <400 mg/dL at day +100 was associated with significantly increased incidence of CS-CMV-i, suggesting that such patients may require LTV extended beyond day +100 [65].

Several CMV-specific immune monitoring assays have been described in allo-HSCT setting, some not constantly reliable or difficult to replicate, whereas other are more standardized [56,62,81–84], all showing capability to predict recurrent and/or late CMV reactivation. However, a prospective study applying a risk stratification models based on the monitoring of CMV-specific cell-mediated immunity to LTV prophylaxis is still lacking.

## 8. Letermovir Use in Our Internal Policy

In 2019, we introduced LTV prophylaxis for high-risk patients defined as CMV positive patients irrespective of donor type, the presence of GVHD requiring steroids and GVHD prophylaxis. Patients received LTV as primary prophylaxis 480 mg/day or 240 mg/day for patients treated with cyclosporine. LTV prophylaxis was given from day +7 until

day +100. Blood CMV-DNA load is monitored using CMV PCR twice weekly from the start of preparative regimen through day +60. From day +60, patients were monitored at each planned visit until day +120 or discontinuation of systemic immunosuppressive therapy.

LTV does not inhibit viral DNA polymerase but rather the late stage of CMV replication, targeting the terminase complex: as a result, long DNA concatemers are not cleaved into single viral subunit resulting in non-infectious DNA fragments, expression of abortive infection (blips) [85]. According to these considerations, potential overinterpretation of PCR results during LTV prophylaxis might be possible. In order to discriminate between abortive infection and infectious CMV-DNAemia, we analyzed CMV viremia trough shell vial assay in case of blips.

Before 2019, a pre-emptive approach was adopted with either ganciclovir, valganciclovir or foscarnet in case of CMV-DNAemia greater than 4.000 U/mL.

In our experience, LTV has strongly reduced CMV reactivation during its use until day 100. In our cohort of 304 allotransplants from 2013 to 2022 at the Stem Cell Transplant Center AOU Città della Salute e della Scienza of Torino, CS-CMV-i at 3 months was 1.3% in LTV group vs. 13.9% in PET group; however, this difference drops after LTV discontinuation, with a 12-months CS-CMV-i of 19.4% vs. 16.1%, respectively (Table 2).

**Table 2.** CS-CMV-i at 1, 3, 6 and 12 months at our center.

| CS-CMV-I | 1 Month | 3 Months | 6 Months | 12 Months |
|---|---|---|---|---|
| Whole cohort | 2.3% | 10.5% | 16.2% | 16.9% |
| LTV group | 0% | 1.3% | 18.1% | 19.4% |
| PET group | 3.1% | 13.9% | 15.7% | 16.1% |

## 9. Conclusions

Primary prophylaxis with letermovir significantly reduces CMV infection in allo-HSCT recipients, potentially leading to a reduction of mortality. This clinical benefit has been reproduced in various real-word studies, overwhelming standard PET. Given the low toxicity profile, letermovir has also been evaluated as secondary prophylaxis and therapy with promising results. Nevertheless, unanswered questions still remain:

(1) The reactivation after +100 days is a remarkable issue; the results of the study with prophylactic letermovir from day +100 to +200 (Clinicaltrials.gov: NCT03930615) are eagerly awaited and might be a useful guide to establish an optimal duration of CMV prophylaxis.
(2) Letermovir might have a low genetic barrier to resistance, potentially leading to a rapid evolution of viral mutations [86].
(3) Blips may be found in up to 30% of patients receiving prophylactic letermovir: additional tools are necessary to discriminate abortive infection from active CMV infection [87].
(4) The optimization of immunological surveillance may be relevant for an accurate prediction of CMV risk, owing to patient-tailored management [88,89].

To conclude, letermovir may be considered as one of the major advances in the prevention of CMV infection in patients receiving allotransplants. Potential alternatives to the current strategies based on antiviral drugs are becoming defined and include cellular and immune therapies.

**Author Contributions:** Conceptualization, A.B., C.D., I.D. and L.G.; methodology, A.B. and L.G.; writing—original draft preparation, A.B., D.S., J.G. and L.G.; writing—review and editing, A.B., D.S., J.G. and L.G.; supervision, A.B., C.D., D.S., I.D., J.G. and L.G. All authors have read and agreed to the published version of the manuscript.

**Funding:** This research received no external funding.

**Institutional Review Board Statement:** The study was conducted in accordance with the Declaration of Helsinki and it was approved by the Ethics Committee of Comitato Etico Interaziendale (C.E.I.) (Project identification code: IT10771180014).

**Informed Consent Statement:** Informed consent was obtained from all subjects involved in the study.

**Data Availability Statement:** The data presented in this study are available on request from the corresponding author. The data are not publicly available due to privacy restrictions.

**Conflicts of Interest:** A.B. and L.G received honoraria from Merck Sharp & Dohme (MSD).

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
