# Peer review of "Use of Letermovir for CMV Prophylaxis after Allogeneic Hematopoietic Stem Cell Transplantation: Review of the Literature and Single-Center Real-Life Experience"

_hemato, doi:10.3390/hemato4020013_

Round 1
Reviewer 1 Report
The article by Gill et al is well written and has merit but will require minor revision to be accepted for publications.
1.There needs to be a correction of grammer in some sentences.
2. Para 1, line 28. GVHD predisposes to CMV reactivation due to activation of donor T cells, so the sentence “ tissue invasive CMV disease predisposes to GVHD” is incorrect. Please correct.
3. Paragraph 2, line 40. The point number 5 is due to ATG and PTCy, so please omit ATG, PTCy from point 1 and mention in point 5.
4. change foscavir to Foscarnet (right name) in line 43.
5. please add data that CMV reactivation has lead to reduced relapses in some studies (PMID: 36824620)
6. Clinicaltrials.gov: NCT03930615. This study was presented at the TCT meeting, Orlando 15 Feb 2023 as oral abstract. Request authors to include initial analysis. The results recommended continuation till Day+200.
7. In the subheading 8, can the authors enumerate there incidence of CS-CMVi or CMV positivity along with a table. That would be useful to the readers. The authors impression on efficacy of the LTV prophylaxis.
Reviewer 2 Report
Comments to the Author
In this review, Jessica Gill and colleagues reviewed the present role of letermovir used as prophylaxis, secondary prophylaxis or therapy post allo-HSCT and discuss some possible future applications of the drug. Overall, this paper provides a good overview of recent research on letermovir. However, the structure of the article needs to be further improved, and some errors need to be corrected.
Major comments:
Q1:The title of this review is “A single center experience with the use of letermovir for CMV prophylaxis after allogeneic hematopoietic stem cell transplantation and review of the literature”. However, only CMV management in author’s center was discussed in the section of “Letermovir use in our internal policy”. I think this section can be delected, or can you put detail data of experience with the use of letermovir for CMV prophylaxis in your center?
Q2:Immunotherapy with donor or third-party derived CMV-specific T cells to restore CMV-specific immunity offers an attractive alternative strategy to conventional anti-CMV therapeutics. It is better to add this treatment strategy in the Introduction.
Q3: In line 51-52, the manuscript mentions that “DNAemia can be detected in blood by different methods, either with CMV antigenemia or more frequently with PCR”. According to Definitions of Cytomegalovirus Infection and Disease in Transplant Patients for Use in Clinical Trials (CID 2017:64) , DNAemia is defifined as the detection of CMV DNA in samples of plasma, serum, whole blood, or isolated PBLs, or in buffy coat specimens.
Q4: Language logic and ideographic were not so unclear, eg Line 35-41.
Q5: It is better to use more charts to summarize literatures to make the review much more clearer.
Round 2
Reviewer 2 Report
Accept in present form